# Macular Choroidal Thickness in Keratoconus: Systematic Review and Meta-Analysis of Current Evidence

**DOI:** 10.3390/diagnostics15182394

**Published:** 2025-09-19

**Authors:** Dimitrios Kazantzis, Genovefa Machairoudia, Panagiotis Theodossiadis, Irini Chatziralli

**Affiliations:** 12nd Department of Ophthalmology, National and Kapodistrian University of Athens, Attikon University Hospital, 12462 Athens, Greece; genevievemach@icloud.com (G.M.); patheo@med.uoa.gr (P.T.); eirchat@yahoo.gr (I.C.); 2Moorfields Eye Hospital NHS Foundation Trust, London EC1V 2PD, UK

**Keywords:** keratoconus, retina, choroidal thickness, optical coherence tomography

## Abstract

**Background/Objectives**: This study aimed to investigate changes and synthesize the existing evidence in macular choroidal thickness in patients with keratoconus (KC) compared to healthy controls, utilizing optical coherence tomography (OCT). **Methods**: PubMed and Scopus databases were systematically searched for published articles comparing choroidal thickness between patients with KC and healthy controls. The Mean Difference (MD) with 95% confidence interval (CI) was computed to compare continuous variables. Our study was registered with PROSPERO with registration ID: CRD42024605227. Revman 5.4 was used for the analysis. **Results** 10 studies were included in the analysis. Subfoveal choroidal thickness was increased in patients with KC compared to controls. (MD = 43.94, 95% CI = 17.36–70.51, *p* = 0.001, I^2^ = 95%). Leave-one-out sensitivity analysis confirmed this finding. **Conclusions**: Our meta-analysis demonstrated that eyes with keratoconus have significantly increased macular choroidal thickness compared to controls. These findings highlight the need for longitudinal studies stratified by disease severity to clarify the role of choroidal changes in keratoconus progression.

## 1. Introduction

The cornea is a transparent, avascular structure forming the anterior surface of the eye and consists of five distinct layers: the epithelium, Bowman’s layer, the stroma, Descemet’s membrane, and the endothelium. The stroma constitutes approximately 90% of corneal thickness and is primarily composed of regularly arranged type I collagen fibrils that maintain corneal strength and transparency. In contrast, Bowman’s layer is a tough, acellular collagen-rich layer that provides additional structural support [1].

Keratoconus (KC) is a progressive ectatic disease of the cornea which results in corneal thinning and steepening with irregular astigmatism that ultimately leads to visual impairment [2]. The etiopathogenesis of the disease is multifactorial and both genetic predisposition and environmental risk factors for KC have been identified [3,4]. Conventionally, KC is considered to affect the anterior segment of the eye. However, an increased body of literature has reported chorioretinal changes in patients with KC implying that there might be posterior segment involvement in patients with KC [5,6,7].

The choroid, situated posterior to the retina, is a highly vascularized tissue composed of several layers including the choriocapillaris and vascular stromal layers, with collagen fibers integral to the vessel walls and interstitial matrix. The organization and distribution of collagen within both the corneal stroma and choroidal stroma are crucial for maintaining ocular structural integrity and may be altered in conditions such as keratoconus, contributing to observed changes in tissue thickness and biomechanics. The choroid is a highly vascularized layer situated between the sclera and retina and plays a vital role in nourishing the outer retina and maintaining ocular homeostasis [8]. Collagen distribution in corneal stroma is affected in patients with KC and as collagen is not only a major component of the corneal stroma but also contributes substantially to the structural integrity of choroidal vessel walls, including the endothelial basal lamina previous studies have hypothesized that collagen alterations might lead to structural choroidal alterations [9]. Optical coherence tomography (OCT) is a non-invasive method that is considered to be the gold standard method of imaging the retina [10]. The advent of enhanced depth imaging in OCT (EDI-OCT) allows the visualization of the choroid in vivo [11]. The choroidal thickness has been found to affected in a number of systemic and ocular diseases [12,13,14,15,16].

A growing body of literature investigates the choroidal thickness in KC patients compared to healthy controls, suggesting the disease process may have more systemic implications than previously recognized. Understanding whether posterior segment alterations, such as changes in the choroid, are present in KC not only broadens our comprehension of the disease’s systemic impact but may also open new avenues for diagnosis, monitoring, and therapeutic intervention. By encompassing both anterior and posterior segment findings, clinical management strategies can be more comprehensive, ultimately aiming to preserve vision and improve quality of life in affected individuals.

In view of the above, the aim of our current study was to systematically review and meta-analyze published data to determine whether macular choroidal thickness differs in patients with keratoconus compared to a control population.

## 2. Materials and Methods

The PUBMED and Scopus databases were searched from inception until the 10 November 2024. The following search was performed: (Keratoconus) AND (Choroidal thickness OR choroid). The titles and abstracts of all the studies recovered by the search were independently screened by two authors (DK and GM) and those found to be eligible were retrieved for a full text assessment. The inclusion criteria were (1) case–control or cross-sectional studies measuring choroidal thickness in the macular region in patients with KC and a control population, (2) diagnosis of KC, and (3) employing OCT to measure choroidal thickness. In addition to electronic searches of PubMed and Scopus, we performed manual searches and screened regional and specialty ophthalmology journals to identify relevant studies that may not be indexed in major databases, ensuring a more comprehensive and inclusive collection of eligible literature. Studies were excluded if they were not published in English language. In addition to PubMed and Scopus, we performed supplementary manual searches to identify potentially relevant studies not indexed in these databases. These included (i) hand-searching the table of contents of selected ophthalmology journals, (ii) screening the reference lists of included studies and relevant reviews, (iii) reviewing conference proceedings and institutional repositories, and (iv) targeted searches in Google Scholar. All records identified through these supplementary strategies were logged and screened using the same predefined eligibility criteria as the database records. All data analyzed were derived from previously published studies and were handled in full compliance with ethical guidelines for systematic reviews. Data extraction forms and quality assessment sheets were stored securely and are available from the corresponding author upon reasonable request. Discrepancies in screening, data extraction, or quality assessment between reviewers (DK and GM) were resolved by consensus, with arbitration by a third author if necessary. No funding or sponsorship influenced the design, execution, or reporting of this meta-analysis. The authors declare no conflicts of interest relevant to the content of this article. Our study was registered with PROSPERO with registration ID: CRD42024605227. The present systematic review and meta-analysis followed the PRISMA guidelines [17]. The prespecified measures of interest were the subfoveal choroidal thickness and the thickness in other macular quadrants. This information was documented by two authors separately in excel sheets. The mean values and the standard deviation of the choroidal thickness were captured, and the following data were extracted from the included studies: first author, year of publication, country or region, study design, eyes of patients with KC and controls, the age of the patients and the controls, the OCT device used the thinnest corneal thickness and the maximal corneal curvature Kmax. The methodological quality of the included studies was assessed using the Newcastle–Ottawa score (NOS), which evaluates the selection and comparability of cases and controls, as well as the ascertainment of the exposure and the non-response rate among the study groups using a scale with a maximum of 9 points [18]. Two authors graded this scale separately.

The descriptive statistics were described as mean ± SD. Continuous outcomes were pooled calculating the mean difference (MD) with 95% confidence interval (CI) as the summary statistic using the inverse-variance random-effects method. When the mean and standard deviation were not available, they were derived from sample size, median and range based on a method previously described by Wan et al. [19]. Statistical analysis was performed using the Review Manager 5.3 software (Copenhagen: The Nordic Cochrane Centre, The Cochrane Collaboration, 2014). Confidence intervals were set at 95%. Inter-study heterogeneity was evaluated by the inconsistency index (I^2^) with values of 25%, 50%, 75% considered indicative of low, moderate, and high heterogeneity, respectively. Subgroup analysis by region or OCT device used was undertaken if possible. Publication bias was evaluated with a visual inspection of funnel plots. Sensitivity analysis was performed by removing one study at a time to confirm that the results were not driven by a single study.

## 3. Results

### 3.1. Study Selection

10 studies met the inclusion and exclusion criteria to be included in the study [20,21,22,23,24,25,26,27,28,29]. Figure 1 describes the selection process. A total of 115 records were identified through database searches in PubMed and Scopus and an additional 12 were identified through manual searches, with 36 duplicates removed prior to screening. Of the remaining 91 records screened by title and abstract, 46 records were excluded. Full-text articles were retrieved for 45 reports, all of which were successfully obtained. After full-text assessment, 35 reports were excluded for the following reasons: insufficient data (*n* = 29), cases and controls not age-matched (*n* = 3), and study design ineligible (*n* = 3). Ten studies met all inclusion criteria and were included in the qualitative and quantitative synthesis. Among the records identified through supplementary searching, one article (Fahmy et al. [24]) was retrieved via Google Scholar. This study met all prespecified eligibility criteria and was included in the final analysis. The studies of Gutierrez-Bonet, Hashemian et al. and Mehrabadi et al. were excluded from the analysis because the cases and controls were not age matched [30,31,32]. The study of Ballasteroz-Sanchez was also excluded from the analysis since it included patients who received cross-linking treatment for keratoconus within 6 months of postoperative evaluation [33].

### 3.2. Study Characteristics

Table 1 presents the characteristics of the included studies. All studies were hospital-based studies, 9 of them were conducted in Europe/Mediterranean Area (5 in Turkey, 2 in Italy, 1 in Spain and 1 in Portugal) and 1 in Saudi Arabia. Most studies employed Heidelberg Spectralis OCT for the measurement of the choroidal thickness, but a few used other devices such as Topcon or Optovue. Most studies reported corneal parameters for the study population such as the thinnest corneal thickness and the maximum keratometry (Kmax), presented in Table 1. All studies reported the subfoveal choroidal thickness, while 5 studies reported the macular choroidal thickness in the nasal and temporal regions 1500 μm to the fovea.

### 3.3. Quality Assessment of the Included Studies

Table 2 presents the NOS for the included studies. Cases were explicitly defined and drawn from a typical clinical setting in all studies. The controls were selected from a hospital-based setting and therefore bias could be introduced and were clearly defined in all studies and had no history of keratoconus. All cases and controls were matched in age but not in other risk factors that could influence the choroidal thickness such as the refractive status of the eye. The choroidal thickness was assessed objectively for patients with keratoconus and controls using the same method. The non-response rate and lost to follow up status for the participants was not clearly stated in any of the studies. The above are summarized in Table 2.

### 3.4. Meta-Analysis of Subfoveal Choroidal Thickness

The subfoveal choroidal thickness was increased in patients with keratoconus compared to control patients (MD = 43.94, 95% CI = 17.36–70.51, *p* = 0.001, I^2^ = 95%, Figure 2). Leave-one-out sensitivity analysis confirmed this finding showing that our results our robust (Table 3). An increase in choroidal thickness was observed in the nasal region 1500 μm from the fovea (MD = 23.91, 95% CI = 6.76–41.071, *p* = 0.006, I^2^ = 77%), whereas no significant difference was found in the temporal region 1500 μm to the fovea (MD = 13.00, 95% CI = −8.60–34.59, *p* = 0.24, I^2^ = 83%, Figure 3); this suggests a possible asymmetric pattern of choroidal thickening in keratoconus Visual inspection of the funnel plot presented in Appendix A did not reveal evidence of publication bias.

### 3.5. Certainty of Evidence

Most studies demonstrated consistent findings favoring increased thickness in KC, but limitations included moderate heterogeneity and some methodological constraints (e.g., device variability, incomplete control of confounders). There is lower certainty in the analysis for the nasal and temporal region to the subfoveal choroidal thickness.

### 3.6. Descriptive Evidence of Keratoconus Severity and Choroidal Thickness

Akkaya et al. found that the central corneal thickness and was negatively correlated with the subfoveal choroidal thickness, but subgroup analysis showed no significant difference between mild and severe KC [20]. Yilmaz et al. reported no correlation between anterior segment parameters and choroidal thickness, while Bilgin et al. likewise found no significant correlation between the thinnest corneal thickness and the subfoveal choroidal thickness [21,23]. In contrast, Aydemir et al. observed that subfoveal choroidal thickness increased with KC severity, and Fahmy et al. noted a positive correlation between corneal curvature and subfoveal choroidal thickness [24,26]. Pierro et al. reported no significant differences across KC severity groups [27].

## 4. Discussion

The present study demonstrates that patients with keratoconus (KC) exhibit significantly increased choroidal thickness in the macular region compared to age-matched controls. This finding contributes to the mounting body of evidence suggesting that KC may involve not just the anterior corneal segment, but also structural alterations in the posterior segment of the eye. Such observations have important implications for our understanding of the pathophysiology, clinical assessment, and potential management strategies of this complex ectatic disorder.

The underlying mechanisms responsible for increased choroidal thickness in KC remain incompletely understood but likely involve a multifaceted interplay between genetic, biochemical, and inflammatory factors. In KC, changes in the collagen production and distribution might be responsible for the structural changes in the choroid. Previous studies have found an abnormal distribution of collagen fibrillar mass resulting in decreased corneal resistance [34]. Moreover, the expression of collagen in the corneal epithelium and stroma has been found to be defective. The corneal stroma shows altered collagen organization, with type I collagen predominating overall and type XII collagen localized primarily in the anterior stroma; these differential changes in collagen subtypes contribute to biomechanical weakening in both anterior and posterior stromal layers and may have parallels in choroidal collagen remodeling associated with increased choroidal thickness [35,36]. Impaired collagen production and distribution in keratoconus might be reflected in the choroid and manifest with an increased choroidal thickness. Another important finding that might explain increased choroidal thickness in KC is the increased activity of proteoglycans (PG) in KC. Akhtar et al. found increased PG in keratoconic corneas [37]. PG are osmotically active molecules that attract water in the choroidal lacunae and therefore increase the choroidal thickness [8].

Keratoconus has been traditionally considered to be a non-inflammatory disease. However, recent findings suggest that inflammation might be part of the pathophysiology of this condition. Inflammatory cytokines such as IL-6 and TNF-a have been found to be increased in the tears of patients with KC compared to controls [38,39]. These inflammation-associated mediators could foster vascular dysplasia, choroidal hyperpermeability, and increased stromal cellularity, potentially mimicking the mechanisms implicated in other pachychoroid spectrum diseases (e.g., central serous chorioretinopathy). It is possible that a subclinical choroidal inflammatory response, driven by a systemic or local shift in cytokine balance, fosters increased choroidal vascularity and stromal edema, ultimately resulting in vascular dilatation and stromal infiltration secondary to inflammation might explain the increased choroidal thickness in KC. The current findings also highlight possible systemic links, as increased choroidal thickness has been found in systemic connective tissue disorders commonly associated with KC, such as Ehlers-Danlos syndrome and Down syndrome [40,41,42]. This raises intriguing questions about whether similar pathogenic pathways may be at play across ocular and systemic tissues, emphasizing the need for interdisciplinary research in this area.

The clinical relevance of the increased choroidal thickness in KC is not yet determined. Feo et al. reported that 10 out of 56 eyes with KC had signs of pachychoroid pigment epitheliopathy [29], a precursor of pachychoroid spectrum of diseases. Previous case series have also described a coexistence of keratoconus with central serous retinopathy, a condition in the spectrum of pachychoroid [43]. While visual function in KC is traditionally attributed to corneal irregularities, posterior segment changes may also play an underappreciated role. The identification of pachychoroid pigment epitheliopathy and co-existing central serous retinopathy in some KC cases suggests that careful posterior segment evaluation may be warranted in selected patients. Future studies should explore whether choroidal thickening in KC confers additional risk for secondary retinal complications, or whether it could serve as a biomarker for disease severity or progression. Therefore, future research should move beyond simple quantitative measurements and instead employ advanced imaging modalities—such as EDI-OCT, swept-source OCT, or OCT angiography—to examine the detailed architecture and microvasculature of the choroid in keratoconus patients. These studies should aim to determine whether qualitative features characteristic of the pachychoroid spectrum, including choroidal vascular hyperpermeability, focal or diffuse RPE changes, and the presence of pachyvessels, are present in this population. Furthermore, establishing standard protocols for morphological assessment will be vital for comparing findings across studies and populations.

Although the evidence base is limited and inconsistent, several studies included in our review explored potential associations between KC severity and choroidal thickness. Some reported positive correlations with parameters such as Kmax or corneal curvature, suggesting a trend toward increasing choroidal thickening with greater disease severity, while others found no significant association. Taken together, these heterogeneous findings highlight both the biological plausibility and the current uncertainty surrounding posterior segment involvement in KC progression. Standardized, longitudinal studies with severity-stratified reporting are needed to clarify whether choroidal changes represent a marker of disease progression or reflect secondary factors such as refractive error and axial length.

Table 4 summarizes the key pathophysiology mechanisms for increased choroidal thickness including collagen dysregulation, hallmarked by abnormal collagen assembly and turnover in the cornea which may extend beyond the anterior segment of the eye to the choroid. Accumulation of proteoglycans–osmotic molecules that attract water- may further contribute to swelling in the choroid of patients with keratoconus. Additionally, subclinical inflammation as suggested by increased cytokine levels in patients with keratoconus might lead to a choroidal stromal remodeling in keratoconus. Finally, the association of keratoconus with systemic connective tissue disorders such as Ehlers-Danlos and Down syndromes—conditions also marked by choroidal changes—adds further support to the notion that these ocular changes may reflect a broader systemic connective tissue disorder rather than a purely localized process.

Our study has several limitations that should be considered when interpreting the results. Firstly, the majority of included studies focused exclusively on measuring subfoveal choroidal thickness, without assessing other choroidal quadrants or regions. This lack of comprehensive regional data restricted our ability to analyze or pool information for choroidal thickness beyond the subfoveal area, potentially overlooking localized or quadrant-specific variations in choroidal involvement among keratoconus patients. Secondly, there was variability in the types of OCT devices and imaging protocols used across studies. Not all studies employed the same OCT model or acquisition parameters, which could account for measurement discrepancies and contribute to the high statistical heterogeneity observed in our meta-analyses. Device-dependent differences in segmentation algorithms and depth penetration may lead to systematic biases, underscoring the need for standardized imaging methodologies in future research. Moreover, important confounding factors and additional parameters that may influence choroidal thickness—such as refractive error, axial length, intraocular pressure, systemic diseases, or medication use—were not uniformly reported or controlled for among the included studies. The absence of these data limits our capacity to adjust for potential confounders, which could have introduced bias or obscured true associations. To mitigate one key source of heterogeneity, we included only studies with age-matched controls as increased age has been found to have an effect and lead to decreased choroidal thickness [44]. However, even with this precaution, other unmeasured variables may still exist. Furthermore, choroidal thickness exhibits significant diurnal variation, typically increasing during the night with peak thickness occurring in the early morning hours and decreasing throughout the day, which underscores the importance of standardized imaging times in studies assessing choroidal thickness to minimize potential confounding effects related to circadian fluctuations [45]. Finally, most of the studies incorporated into our analysis were small, hospital-based case–control studies. Such study designs are susceptible to selection bias, often lack longitudinal follow-up, and are limited in their ability to establish causal relationships or generalize results to wider populations. Larger, prospective, population-based studies are needed to confirm our findings and clarify the clinical relevance of increased choroidal thickness in keratoconus.

## 5. Conclusions

In conclusion, we found that the choroid is increased in eyes with KC compared to controls in the subfoveal region. Future research is warranted to confirm this finding with well-designed studies. KC might involve the posterior segment and be linked with pachychoroid spectrum of diseases.

## Figures and Tables

**Figure 1 diagnostics-15-02394-f001:**
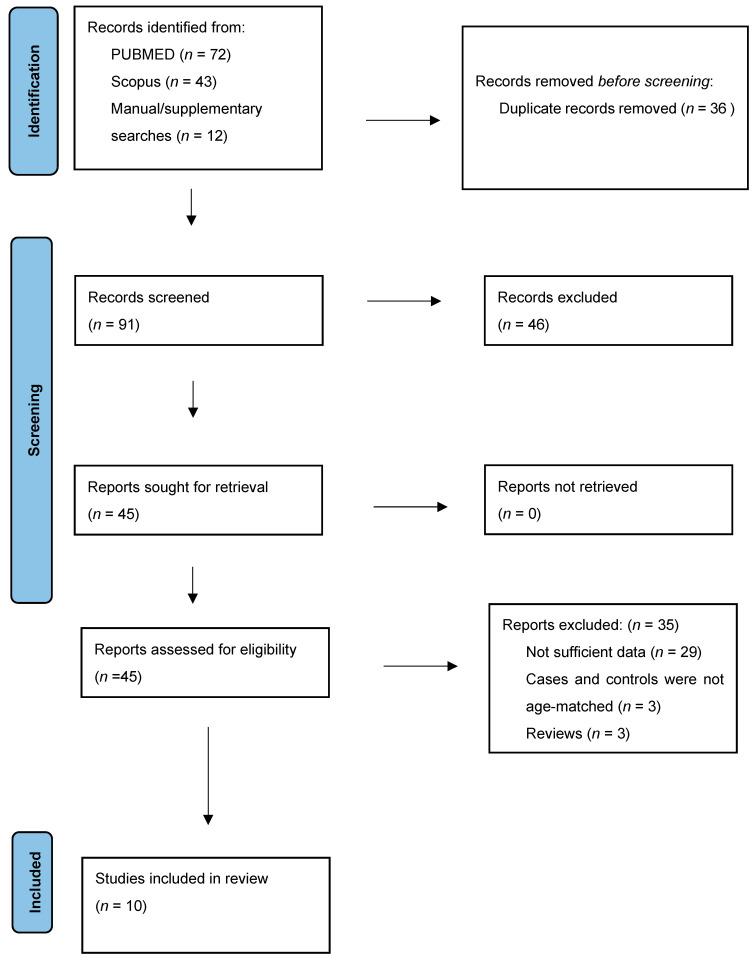
Flowchart of the included studies.

**Figure 2 diagnostics-15-02394-f002:**
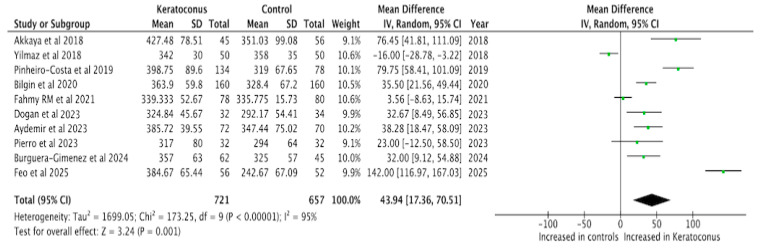
Forest plot of the mean difference in the subfoveal choroidal thickness between patients with keratoconus and controls [20,21,22,23,24,25,26,27,28,29].

**Figure 3 diagnostics-15-02394-f003:**
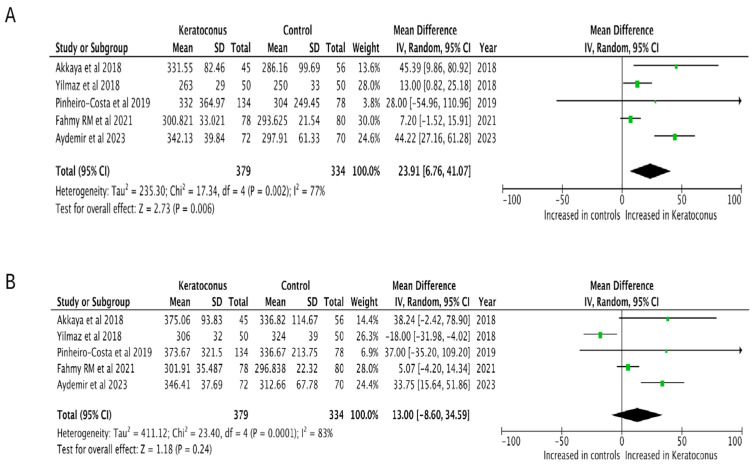
Forest plot of the mean difference in the (**A**) nasal and (**B**) temporal choroidal thickness between patients with keratoconus and controls [20,21,22,24,26].

**Table 1 diagnostics-15-02394-t001:** Characteristics of the included studies.

Author/Ref.	Year	Region	Study Design	No of Eyes of Patients/Controls	Age of Patients/Controls (Years)	Thinnest Corneal Thickness (μm)	Kmax (D)	OCT Device
Akkaya [20]	2018	Turkey	Prospective cross-sectional	45/56	24.5 ± 7.2/22.5 ± 7.4	na	na	Spectralis OCT, Heidelberg Engineering
Yilmaz [21]	2018	Turkey	Cross-sectional	50/50	12.4 ± 1.9/12.0 ± 2.1	456 ± 57	57.45 ± 11.16	Spectralis OCT, Heidelberg Engineering
Pinheiro-Costa [22]	2019	Portugal	Case–control	74/39	23.01 ± 4.68/22.40 ± 5.77	456.66 ± 51.91	56.49 ± 7.83	Spectralis OCT, Heidelberg Engineering
Bilgin [23]	2020	Turkey	Cross-sectional	80/80	18.9 ± 6.9/19 ± 5.6	449.7 ± 3.2	na	Spectralis OCT, Heidelberg Engineering
Fahmy [24]	2021	Saudi Arabia	Case–control	78/80	29.60 ± 7.405 and 27.55 ± 7.207 for males and females/22.625 ± 4.87 and 23.0 ± 5.416 in controls	na	na	Spectralis OCT, Heidelberg Engineering
Dogan [25]	2023	Turkey	Case–control	32/24	28.50 ± 11.48 /30.63 ± 8.17	464.65 ± 42.41	54.93 ± 3.92	OCT Optovue
Aydemir [26]	2023	Turkey	Prospective cross-sectional	72/113	24.44 ± 6.35/21.55 ± 5.30	na	49.82 ± 4.47	Spectralis OCT, Heidelberg Engineering
Pierro [27]	2023	Italy	Observational case–control	32/32	26.92 ± 9.6/26.66 ± 1.8	495 ± 32	49.77 ± 4	Topcon Triton OCT
Burguera-Giménez [28]	2024	Spain	Prospective cross-sectional	62/45	30 ± 12/ 32 ± 9.1	483 ± 39.42	48.40 ± 4.73	DRI-1 OCT, Topcon Medical
Feo [29]	2025	Italy	Retrospective cross-sectional	56/62	35.2 ± 13.2/34.6 ± 15.0	466 ± 47	50.7 ± 4.8	Spectralis OCT, Heidelberg Engineering

OCT: Optical coherence tomography, na: not available, Kmax: Maximum corneal curvature, μm: micrometers D: diopters.

**Table 2 diagnostics-15-02394-t002:** Newcastle–Ottawa scale scores of the included studies.

Study (Year)/Ref.	Selection				Comparability		Exposure			Total
	Case Definition	Representativeness of the Cases	Selection of Controls	Definition of Controls	On Age	On Other Risk Factors	Assessment of Exposure	Same Method of Assessment in Cases and Controls	Non-Response Rate	
Akkaya (2018) [20]	1	1	0	1	1	1	1	1	0	7
Yilmaz (2018) [21]	1	1	0	1	1	0	1	1	0	6
Pinheiro-Costa (2019) [22]	1	1	0	1	1	0	1	1	0	6
Bilgin (2020) [23]	1	1	0	1	1	0	1	1	0	6
Fahmy (2021) [24]	1	1	0	1	1	0	1	1	0	6
Dogan (2023) [25]	1	1	0	1	1	0	1	1	0	6
Aydemir (2023) [26]	1	1	0	1	1	1	1	1	0	7
Pierro (2023) [27]	1	1	0	1	1	1	1	1	0	7
Burguera-Giménez (2024) [28]	1	1	0	1	1	1	1	1	0	7
Feo (2025) [29]	1	1	0	1	1	1	1	1	0	7

**Table 3 diagnostics-15-02394-t003:** Results of the leave-one-out sensitivity analysis.

Study (Year)/Ref.	MD	95% CI	*p*-Value	I^2^
Akkaya (2018) [20]	40.66	12.85–68.47	0.004	95%
Yilmaz (2018) [21]	50.94	24.68–77.20	0.0001	93%
Pinheiro-Costa (2019) [22]	39.85	12.58–67.12	0.004	95%
Bilgin (2020) [23]	45.08	14.00–76.16	0.004	95%
Fahmy (2021) [24]	48.80	18.53–79.07	0.002	95%
Dogan (2023) [25]	45.25	16.15–74.34	0.002	95%
Aydemir (2023) [26]	44.67	15.11–74.22	0.003	95%
Pierro (2023) [27]	46.06	17.65–74.49	0.001	95%
Burguera-Giménez (2024) [28]	45.34	16.11–74.57	0.002	95%
Feo (2025) [29]	32.63	11.74–53.51	0.002	91%

MD: mean difference, CI: confidence interval.

**Table 4 diagnostics-15-02394-t004:** Key hypothesized mechanisms for increased choroidal thickness in keratoconus.

Mechanism	Potential Pathologic Contributor	Supporting Evidence
Collagen dysregulation	Impaired corneal expression in choroid	Abnormal corneal collagen distribution in KC in previous studies [33,34,35]
Proteoglycan accumulation	Osmotic choroidal swelling	Increased proteoglycans in keratoconus [36,37]
Subclinical Inflammation	Cytokine-mediated vascular remodeling	Elevated cytokine levels in tears of patients with keratoconus [8,38]
Systemic associations	Shared connective tissue phenotypes	Increased choroidal thickness in patients with Ehler–Danlos and Down syndrome [39,40,41]

KC: keratoconus.

## Data Availability

The data presented in this study are available from the corresponding authors upon reasonable request.

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
