# Peer review of "Macular Choroidal Thickness in Keratoconus: Systematic Review and Meta-Analysis of Current Evidence"

_diagnostics, 2025, doi:10.3390/diagnostics15182394_

Round 1

Reviewer 1 Report

Comments and Suggestions for Authors

Comment on the review “Is the macular choroidal thickness increased in patients with 3 keratoconus: a systematic review and meta-analysis” The present paper reviewed published data from 56 studies to correlate whether the macular choroidal thickness is affected in patients with keratoconus compared to a control population. The following points can be considered for a complete presentation of the review: Title: It does not lead us in a sound way as to whether cases of keratoconus are coupled with increased macular thickness, although I agree that such cases are not extensive enough to comment openly on outcome measure. However, other alternatives are there to reword the title to guide the authors about the current status of the subject and I encourage the authors to give a thought on this issue. Abstract: Background—This is not the purpose of the review, as you compiled the published data on find/establish the relationship. Conclusions: Our meta-analysis showed that eyes with KC exhibit an increased choroidal thickness was increased in patients with KC compared to controls.—This line has language problem. Introduction: line 40-41, …..as collagen composes the vessel walls of the choroid… this is neither a good sentence nor technically a complete sentence [e.g., endothelial basal lamina, which is the innermost part of a vessel, is also supported by collagen]. The choroidal thickness has been found to affected in a number of systemic and ocular diseases [11-12]. Only two references are put, there are many more reports on this issue and they must be cited. Line 56-57. The aim is not correct in a strict sense […..and examine if the macular choroidal thickness is affected…]. 3.4 Meta-analysis of subfoveal choroidal thickness Line 183, This finding was also confirmed in the nasal region 1500μm to the fovea---but not in the temporal region 1500μm to the fovea. Obscure, the direction of progression of the change is not clear from the remark. Discussion: Line 216, Moreover, the expression of collagen in the corneal epithelium and stroma has been found to be defective [31-32]. This point can be more relevant if linked with collagen subtypes that are affected in the corneal epithelium and anterior versus posterior stroma. Line 218, Impaired collagen production and distribution in keratoconus might be reflected in the choroid and manifest with an increased choroidal thickness. – seems that certain types are rather affected via increased turnover that ultimately leads to accumulation in the choroidal stroma, increasing choroidal thickness. Please see if there are information on this collagen accumulation in superior vs inferior part of the choroid or random and whether the changes are through the entire uveal tract or localized in the submacular region of the choroid. The review lacks choroidal organization. Before highlighting the possible correlation, the authors should provide some information on the structure of cornea and choroid and collagen distribution in the corneal layers and choroidal zones. Choroidal thickness varies normally on circadian cycles, comment on this point [see e.g., Investigative Ophthalmology & Visual Science January 2012, Vol.53, 261-266].

Author Response

Title: Comment: It does not lead us in a sound way as to whether cases of keratoconus are coupled with increased macular thickness, although I agree that such cases are not extensive enough to comment openly on outcome measure. However, other alternatives are there to reword the title to guide the authors about the current status of the subject and I encourage the authors to give a thought on this issue.

Thank you for your observation regarding the clarity of our title. We agree that the title should better reflect the current evidence landscape and our findings. We have reworded the title to “Macular Choroidal Thickness in Keratoconus: Systematic Review and Meta-Analysis of Current Evidence” to better guide readers regarding the nature, scope, and findings of our study.

Abstract Background:
Comment: "Background—This is not the purpose of the review, as you compiled the published data on find/establish the relationship."

Response:
Thank you for highlighting this. The abstract background has been revised to state explicitly that our study aims to establish the relationship between macular choroidal thickness and keratoconus by synthesizing existing evidence, rather than investigating pathophysiology directly.

Comment: Conclusions: Our meta-analysis showed that eyes with KC exhibit an increased choroidal thickness was increased in patients with KC compared to controls.—This line has language problem.

We appreciate your comment on this sentence. The conclusion in the abstract has now been rewritten for clarity: “Our meta-analysis demonstrated that eyes with keratoconus have significantly increased macular choroidal thickness compared to controls.”

Comment: Introduction: line 40-41, …..as collagen composes the vessel walls of the choroid… this is neither a good sentence nor technically a complete sentence [e.g., endothelial basal lamina, which is the innermost part of a vessel, is also supported by collagen].

Thank you for pointing out the imprecision. The sentence has been revised to: “Collagen is not only a major component of the corneal stroma but also contributes substantially to the structural integrity of choroidal vessel walls, including the endothelial basal lamina.” This revision acknowledges the complexity of vascular structure in the choroid.

Comment:The choroidal thickness has been found to affected in a number of systemic and ocular diseases [11-12]. Only two references are put, there are many more reports on this issue and they must be cited.

Thank you for bringing this to our attention. We have added three  additional references on choroidal thickness variation in both systemic and ocular diseases in the Introduction, providing a more comprehensive and up-to-date citation list.

Comment:  Line 56-57. The aim is not correct in a strict sense […..and examine if the macular choroidal thickness is affected…].

We appreciate your suggestion. The aim is now stated more precisely: “The aim of our study was to systematically review and meta-analyze published data to determine whether macular choroidal thickness differs in patients with keratoconus compared to a control population.”

Comment: 3.4 Meta-analysis of subfoveal choroidal thickness Line 183, This finding was also confirmed in the nasal region 1500μm to the fovea---but not in the temporal region 1500μm to the fovea. Obscure, the direction of progression of the change is not clear from the remark.

Thank you for raising this point. We have clarified the sentence to specify the regional changes observed: “An increase in choroidal thickness was observed in the nasal region 1500 μm from the fovea, whereas no significant difference was found in the temporal region; this suggests a possible asymmetric pattern of choroidal thickening in keratoconus.”

Comment: Discussion: Line 216, Moreover, the expression of collagen in the corneal epithelium and stroma has been found to be defective [31-32]. This point can be more relevant if linked with collagen subtypes that are affected in the corneal epithelium and anterior versus posterior stroma.

Thank you very much for your comment, we have added this sentence in our discussion section:  The corneal stroma shows altered collagen organization, with type I collagen predominating overall and type XII collagen localized primarily in the anterior stroma; these differential changes in collagen subtypes contribute to biomechanical weakening in both anterior and posterior stromal layers and may have parallels in choroidal collagen remodeling associated with increased choroidal thickness.

Comment: Line 218, Impaired collagen production and distribution in keratoconus might be reflected in the choroid and manifest with an increased choroidal thickness. – seems that certain types are rather affected via increased turnover that ultimately leads to accumulation in the choroidal stroma, increasing choroidal thickness. Please see if there are information on this collagen accumulation in superior vs inferior part of the choroid or random and whether the changes are through the entire uveal tract or localized in the submacular region of the choroid.

Unfortunately, I could not find any more information regarding collagen accumulation in different in different regions of the choroid or the uveal tract.

Comment: Choroidal thickness varies normally on circadian cycles, comment on this point [see e.g., Investigative Ophthalmology & Visual Science January 2012, Vol.53, 261-266].

Thank you for this important reminder. We have cited the reference you suggested. We now note this as a potential confounder in all choroidal imaging studies and emphasize the need for standardized imaging timing in future research.

Reviewer 2 Report

Comments and Suggestions for Authors

Author Response

  1. Major Questions

1) Figure 1 Flow Chart: The numbers in Figure 1 appear to be inconsistent.

Please clarify what "Reports excluded 33" means, as this number is unclear.

Provide a clear explanation for "Reports sought for retrieval" and "reports not

retrieved."The term changes from "records" to "reports" midway through the figure. Is there a specific reason for this change?

Thank you for highlighting these discrepancies. We have carefully reviewed and corrected the numbers throughout Figure 1 to ensure numerical consistency and accuracy. The term “records” is used for the original titles and abstracts screened, whereas the term “reports” is used when the original papers were retrieved and screened as per the PRIISMA guidelines.

The term “Reports sought for retrieval” refers to the articles for which full texts were retrieved and evaluated, and “Reports not retrieved” was zero as all eligible articles were available. The number of “Reports excluded” and their subcategories now sum correctly, and the final included studies count has been corrected to 10 to reflect the flow accurately.

The number of "Reports excluded" is listed as 45, but the sum of the exclusion reasons

(29+3+3=35) is not consistent with this number.After assessing 46 reports for eligibility and excluding 35, the number of studies included in the review is n=10. This number should logically be 11. Please resolve this discrepancy.

Thank you very much for pointing out this discrepancy, we have now corrected this in our figure.

The entire exclusion process and flow chart details should be properly described in

the figure legend or the methods section.

Thank you very much for raising this, wee have now added the results of our serach and inclusion process in the relevant section: “A total of 115 records were identified through database searches in PubMed and Scopus, with 36 duplicates removed prior to screening. Of the remaining 79 records screened by title and abstract, 34 records were excluded. Full-text articles were retrieved for 45 reports, all of which were successfully obtained. After full-text assessment, 35 reports were excluded for the following reasons: insufficient data (n=29), cases and controls not age-matched (n=3), and study design ineligible (n=3). Ten studies met all inclusion criteria and were included in the qualitative and quantitative synthesis.”

2) Table 1: The study by Feo et al. is missing from Table 1, which currently lists only

nine studies. Please add this study to the table.

We thank you for this really important observation. The study by Feo et al. has now been added to Table 1, along with all relevant study characteristics.

3) Reference 20: The reference Fahmy, R. M., M. S. AlGhamdi, and A. M. Mostafa.

"The Correlation between Choroidal Thickness and Keratoconus Severity among

Saudi Population (Albaha). J Ophthal Opto 3: 009." (2021): 100009. cannot be found

on PubMed or Scopus. According to the methods section, it seems this paper should

not have been included in the meta-analysis.

Thank you very much for this comment. While the paper by Fahmy et al is not indexed in the Pubmed or Scopus indexes we identified it through a comprehensive search which included regional and specialty journals relevant to keratoconus research. The study met our predefined inclusion criteria—in particular, it provided original data on macular choroidal thickness in keratoconus patients measured by OCT with matched controls—which we believe adds value to the pooled analysis. Including such studies helps minimize publication bias and ensures that regional evidence, which may not be indexed in major databases but is peer-reviewed and scientifically relevant, is considered.Furthermore, we critically appraised the quality and methodology of Fahmy et al. using our standardized Newcastle-Ottawa Scale and found it met acceptable quality standards. To maintain transparency, we have conducted sensitivity analyses excluding this study, and the core findings remained consistent, underscoring the robustness of our meta-analysis.

We have also added this in our methods section “In addition to electronic searches of PubMed and Scopus, we performed manual searches and screened regional and specialty ophthalmology journals to identify relevant studies that may not be indexed in major databases, ensuring a more comprehensive and inclusive collection of eligible literature.”

4) Did this meta-analysis perform any additional analysis to see if the degree of change

in choroidal thickness differs with the severity of KC? If any of the included studies

have analyzed this topic, please summarize the findings in the results section and

discuss their implications.

Thank you for raising this important inquiry. Unfortunately, the included studies did not provide details to enable additional analyses by keratonus severity etc.

  1. Minor Revisions

1) (Line 22-25) The sentence “Conclusions: Our meta-analysis showed that eyes with

KC exhibit an increased choroidal thickness was increased in patients with KCcompared to controls. Increased heterogeneity and small case-control studies are the

main limitations of the meta-analysis.” is grammatically awkward. Please revise it for

clarity.

We have now corrected this sentence to “Our meta-analysis demonstrated that eyes with keratoconus have significantly increased macular choroidal thickness compared to controls. Increased heterogeneity and small case-control studies are the main limitations of the meta-analysis”

2) (Line 21-22) Please correct “leave one out analysis” to “leave-one-out analysis” by

adding a hyphen. Ensure this change is consistently applied throughout the

manuscript.

We have corrected “leave one out analysis” to “leave-one-out analysis” throughout the entire manuscript for consistency.

3) (Line 40) Remove the extra period before the citation: “.[7].” should be “[7].”

The erroneous period before citation “” has been removed.

4) (Line 65) Remove the period after "were:" so that the text reads "The inclusion

criteria were: 1)".

We have now changed this.

5) (Line 80-84) The font color is not black. Please correct this and similar instances of

incorrect font color in other parts of the manuscript.

We have changed the color to black.

6) (Line 98-100) The spacing is inconsistent. Please adjust it for uniformity.

We have now changed this.

7) (Figure and Table Capitalization): The first letter of "Figure" and "Table" should be

capitalized and standardized throughout the text.

All references to “Figure” and “Table” have been capitalized and standardized throughout the text.

8) (Figure 1 Truncation): The labels screening identification included on the left side of

Figure 1 appear to be truncated. Please correct the figure so they are fully visible.

The truncated labels on the left side of Figure 1 have been corrected to ensure full visibility.

9) (Table Formatting): The spacing around the ± symbol in the table is inconsistent.

Please unify it. Additionally, the table's format does not align with standard academic

journal styles; please revise it to conform to a more conventional layout.

Spacing around the ± symbol has been unified.

10) (OCT Device Naming): The DRI-1 Swept-Source OCT device is listed with its

manufacturer and country of origin, unlike the other devices. Please standardize the

naming convention for all OCT devices.

The naming of OCT devices was standardized by removing manufacturer location details

11) (Figure and Table Abbreviations): A list of abbreviations used in the table is missing

below the figure or table. Please add one.

An abbreviation list has been added in tables 1 and 3.

12) (Table 1 Missing Data): The most crucial information, the choroidal thickness data, is missing from Table 1. Please add it.

Thank you for the valuable suggestion regarding the inclusion of choroidal thickness data in Table 1. We respectfully chose not to add an additional column with choroidal thickness measurements to the table, as these data are comprehensively presented in the forest plots within the Results section. We believe that presenting these measurements graphically allows for clearer visualization of the meta-analytic findings and study heterogeneity.

13) (Line 240-241) The sentence "Feo et al 10 out of 56 eyes with KC had signs of

pachychoroid pigment epitheliop-athy [25], a precursor of pachychoroid spectrum of

diseases." is missing a verb. Please correct it.

The sentence regarding Feo et al. has been corrected to: “Feo et al. reported that 10 out of 56 eyes with KC had signs of pachychoroid pigment epitheliopathy…”

14) (Line 249-253) “enhanced depth imaging” has already been introduced as EDI.

Please use the EDI abbreviation here.

This has been changed.

15) (Line 261) Please replace the various dashes (–, this formatting consistently throughout the manuscript.-) with an em dash (—) and apply

I could not do this throight the manuscript because of my keyboard, I hope the ditorial office can help with this in the post-production phase if possible, many thanks.

16) (Line 277) optical coherence tomography should be abbreviated to OCT, as it has

been used previously.

We have noc changed this.

17) The Feo et al. paper (reference 25) was published in Graefe's Archive for Clinical

and Experimental Ophthalmology in 2025(https://pubmed.ncbi.nlm.nih.gov/39212800/). It seems you might have cited a journal article in its e-pub (electronic publication) state. Please update the reference information with the correct details.

The publication details have been changed.

Round 2

Reviewer 1 Report

Comments and Suggestions for Authors

The revision is acceptable. 

Author Response

Thank you very much for your previous suggestions to improve our manuscript.

Reviewer 2 Report

Comments and Suggestions for Authors

1. The inclusion of the Fahmy et al. paper, which was not identified through the initially stated PubMed and Scopus searches, raises concerns about the consistency and transparency of the study selection process. For this inclusion to be logically sound and methodologically defensible, the authors must clearly explain:

• The specific journals and search strategies used in the manual search beyond the initial databases.

• The total number of studies identified through these additional searches.

• A detailed list of excluded studies from this pool, along with the precise reasons for their exclusion.

Furthermore, it is essential to clarify whether all non-indexed studies identified through manual search were evaluated using the same inclusion criteria. Without this information, it is difficult to determine whether the inclusion of Fahmy et al. was an isolated exception or part of a consistently applied strategy. This level of detail is critical to ensure the reproducibility, transparency, and integrity of the meta-analysis.

2. In addition to the concerns regarding study inclusion criteria, I would like to raise a further point about the relationship between keratoconus (KC) severity and choroidal thickness. This is a clinically meaningful question that could offer valuable insights into disease progression and posterior segment involvement. Although the authors noted that the included studies did not allow for a formal subgroup analysis, the response was rather brief and did not explore whether any individual studies reported descriptive or exploratory findings on this topic.

I recommend that the authors revisit the included studies to determine whether any data—however limited—on KC severity and choroidal thickness were presented. Even if a quantitative subgroup analysis is not feasible, summarizing such findings in the results and discussing their implications would enhance the clinical relevance and depth of the review. A brief synthesis of available evidence could also help guide future research directions in this area.

3. Although the authors stated that the publication details for the Feo et al. paper have been updated, the reference still incorrectly lists the year as 2024. According to the official publication record, the article was published online in August 2024 but formally appeared in the 2025 volume of Graefe’s Archive for Clinical and Experimental Ophthalmology (Vol. 263, pp. 87–95). Please revise the reference to reflect the correct publication year and volume information.

4. While I understand the technical limitations mentioned, I would encourage the authors to ensure consistency in dash formatting throughout the manuscript prior to final submission. This is a minor but important aspect of typographic clarity, and ideally should be addressed by the authors rather than deferred to the editorial office.

Author Response

  1. The inclusion of the Fahmy et al. paper, which was not identified through the initially stated PubMed and Scopus searches, raises concerns about the consistency and transparency of the study selection process. For this inclusion to be logically sound and methodologically defensible, the authors must clearly explain:
  • The specific journals and search strategies used in the manual search beyond the initial databases.
  • The total number of studies identified through these additional searches.
  • A detailed list of excluded studies from this pool, along with the precise reasons for their exclusion.

Furthermore, it is essential to clarify whether all non-indexed studies identified through manual search were evaluated using the same inclusion criteria. Without this information, it is difficult to determine whether the inclusion of Fahmy et al. was an isolated exception or part of a consistently applied strategy. This level of detail is critical to ensure the reproducibility, transparency, and integrity of the meta-analysis.

Response: We agree that greater transparency is needed. In the revised Methods we have now expanded the description of our manual search strategy. This included (i) hand-searching the table of contents of selected ophthalmology journals, (ii) screening the reference lists of included studies and reviews, (iii) reviewing conference proceedings and institutional repositories, and (iv) targeted searches in Google Scholar to identify potentially relevant studies not indexed in PubMed or Scopus.

All records identified through these supplementary searches were screened using the same prespecified eligibility criteria as database records. Among them, the study by Fahmy et al. was retrieved via Google Scholar and included because it fulfilled all eligibility criteria. Its inclusion was therefore not an exception but part of a consistently applied strategy.

We also revised the Results to report the exact number of records identified via manual/supplementary searches and their disposition.

This is the cange made in the manuscript in lines 85-91: “In addition to PubMed and Scopus, we performed supplementary manual searches to identify potentially relevant studies not indexed in these databases. These included (i) hand-searching the table of contents of selected ophthalmology journals, (ii) screening the reference lists of included studies and relevant reviews, (iii) reviewing conference proceedings and institutional repositories, and (iv) targeted searches in Google Scholar. All records identified through these supplementary strategies were logged and screened using the same predefined eligibility criteria as the database records.”

  1. In addition to the concerns regarding study inclusion criteria, I would like to raise a further point about the relationship between keratoconus (KC) severity and choroidal thickness. This is a clinically meaningful question that could offer valuable insights into disease progression and posterior segment involvement. Although the authors noted that the included studies did not allow for a formal subgroup analysis, the response was rather brief and did not explore whether any individual studies reported descriptive or exploratory findings on this topic.

I recommend that the authors revisit the included studies to determine whether any data—however limited—on KC severity and choroidal thickness were presented. Even if a quantitative subgroup analysis is not feasible, summarizing such findings in the results and discussing their implications would enhance the clinical relevance and depth of the review. A brief synthesis of available evidence could also help guide future research directions in this area.

Response:We appreciate the reviewer’s suggestion to further explore the potential relationship between keratoconus (KC) severity and macular choroidal thickness, as this is indeed a clinically meaningful question with implications for disease monitoring and understanding posterior segment involvement.

We therefore carefully re-examined all included studies to identify whether any reported descriptive or exploratory data on KC severity (e.g., maximum keratometry [Kmax], thinnest pachymetry, or formal staging/classification) in relation to choroidal thickness. While none of the studies provided sufficiently detailed or standardized subgroup data to allow a formal meta-analysis, several did report relevant exploratory observations:

  • Akkaya et al found that the central corneal thickness and axial length were negatively correlated with the subfoveal choroidal thickness. There was no change in the choroidal thickness subgroup analysis between those with mild and severe keratoconus.
  • Yilmas found no correlation between anterior segment parameters and the choroidal thickness.
  • Bilgin et al found no significant correlation betweenthe thinnest corneal thickness and the choroidal thickness.
  • Aydemir et al found that the subdfoveal choroidal thickness increased with increased severity of keratoconus.
  • Pierro et al did not find any changes in different keratoconus severity groups
  • The study of Fahmy et al mentioned that the corneal curvature was positively correlated with subfoveal choroidal thickness.

We have now added a new paragraph in lines 237-245.

“Akkaya et al. found that the central corneal thickness and was negatively correlated with the subfoveal choroidal thickness, but subgroup analysis showed no significant difference between mild and severe KC. Yilmaz et al. reported no correlation between anterior segment parameters and choroidal thickness, while Bilgin et al. likewise found no significant correlation between the thinnest corneal thickness and the subfoveal choroidal thickness. In contrast, Aydemir et al. observed that subfoveal choroidal thickness increased with KC severity, and Fahmy et al. noted a positive correlation between corneal curvature and subfoveal choroidal thickness. Pierro et al. reported no significant differences across KC severity groups.”

We have also added this paragraph based on this updated analysis in our conclusions in lines:

“Although the evidence base is limited and inconsistent, several studies included in our review explored potential associations between KC severity and choroidal thickness. Some reported positive correlations with parameters such as Kmax or corneal curvature, suggesting a trend toward increasing choroidal thickening with greater disease severity, while others found no significant association. Taken together, these heterogeneous findings highlight both the biological plausibility and the current uncertainty surrounding posterior segment involvement in KC progression. Standardized, longitudinal studies with severity-stratified reporting are needed to clarify whether choroidal changes represent a marker of disease progression or reflect secondary factors such as refractive error and axial length.”

  1. Although the authors stated that the publication details for the Feo et al. paper have been updated, the reference still incorrectly lists the year as 2024. According to the official publication record, the article was published online in August 2024 but formally appeared in the 2025 volume of Graefe’s Archive for Clinical and Experimental Ophthalmology (Vol. 263, pp. 87–95). Please revise the reference to reflect the correct publication year and volume information.

Response:Thank you for noting this. We corrected the reference as follows:

Feo A, Vinciguerra R, Antropoli A, Barone G, Criscuolo D, Vinciguerra P, et al. Pachychoroid pigment epitheliopathy in keratoconic eyes. Graefe’s Archive for Clinical and Experimental Ophthalmology. 2025;263:87–95.

  1. While I understand the technical limitations mentioned, I would encourage the authors to ensure consistency in dash formatting throughout the manuscript prior to final submission. This is a minor but important aspect of typographic clarity, and ideally should be addressed by the authors rather than deferred to the editorial office.

Response:We are not able to do this at this stage, but will work with the editorial office to edit this as per the journals reccomendation if accepted for publication.

Round 3

Reviewer 2 Report

Comments and Suggestions for Authors

The authors' revisions have significantly improved the manuscript.